# Study on Radiation Shielding Properties of New Barium-Doped Zinc Tellurite Glass

**DOI:** 10.3390/ma15062117

**Published:** 2022-03-13

**Authors:** Shiyu Yin, Hao Wang, Aifeng Li, Zhongjian Ma, Yintong He

**Affiliations:** 1School of Mechanical and Materials Engineering, North China University of Technology, Beijing 100144, China; xizaidaoren@163.com; 2College of Information Science and Enginfeering, Shandong Agricultural University, Taian 271018, China; 3Institute of High Energy Physics, Chinese Academy of Sciences, Beijing 100049, China; mazhj@ihep.ac.cn; 4Innovation & Research Institute of HIRING Technology, Beijing 100074, China; heytong_83@126.com

**Keywords:** radiation-shielding, telluride glass, high-energy rays

## Abstract

This study aimed to investigate the effect of BaF_2_ on the radiation-shielding ability of lead telluride glass. A physical radioactive source was used to estimate the mass attenuation coefficient (μ_m_) of the 60TeO_2_-20PbO-(20-x)ZnO-xBaF_2_ glass system (where x = 1,2,3,5,6,7,9 mol%). We tested the μ_m_ values at seven energies (0.059, 0.081, 0.122, 0.356, 0.662, 1.173, 1.332 MeV). To determine the accuracy of the obtained results, we compared the experimental data with the data calculated using the XCOM software. The experimental values obtained for the selected lead telluride glasses at different concentrations of BaF_2_ are in good agreement with the results of XCOM at all energies. The addition of BaF_2_ increased the μ_m_ value of the sample. At the same time, the half-value layer (HVL), mean free path (MFP), effective atomic number (Z_eff_), and fast neutron removal cross-section (RCS) of the glass were studied. With the increase in the BaF_2_ content, the HVL value and MFP value of the glass decreased, and the Z_eff_ value and RCS increased, indicating that the addition of BaF_2_ enhanced the radiation-shielding performance of the glass.

## 1. Introduction

In recent years, with the continuous development of research or industrial equipment such as industrial X-ray systems, reflex therapy, nuclear power generation, and particle accelerators, reducing ionizing radiation and preventing it from causing damage to workers or the environment has become an important research topic. Therefore, researchers are constantly trying to find new high-quality radiation-shielding materials, with the lowest possible cost to attenuate radiation to a safe and acceptable level [1,2,3,4,5].

Suitable radiation-shielding materials should meet the conditions of environmental protection, durability, transparency, and easy production at the same time. Researchers have tried to study alloys, lead-free concrete, rocks, polymers, and glass in developing radiation shielding materials. Many research groups are currently focusing on different glass systems among the above materials [6,7,8]. This is because glass is simple to prepare and has excellent comprehensive properties such as high transmittance, damage resistance, pressure resistance, and heat resistance. In addition, the density and radiation-shielding ability of glass can be significantly improved by doping various heavy metal elements (such as Bi, Pb, Ba, and Gd). Many research groups have reported on glass of various substrates and their nuclear protective properties in the literature [9,10,11].

In recent years, more and more attention has been paid to the research on the radiation shielding performance of lead telluride glass. Using TeO_2_ as the matrix, Lead telluride glass has a larger density and radiation absorption cross-section. After receiving radiation, their optical and mechanical properties of lead tellurite glass change only slightly, making it an excellent transparent radiation-shielding material. At the same time, lead telluride glass has a high refractive index that other common radiation-shielding materials do not have. For optical components, the higher the refractive index, the thinner the material can become, and the smaller the volume of the entire optical system. In some special working environments, such as space operations and medical equipment requiring radiation shielding ability and small instrument volume, the advantages of high refractive index glass are apparent [12,13,14,15,16,17]. Zinc oxide is added to the glass composition to ensure that the glass has a good structure and thermal stability [18,19]. At the same time, the incorporation of Ba in the glass can improve its optical and radiation shielding capabilities, and the toxicity of barium glass is very low to the human body [20].

The effect of the BaF_2_ additive on a TeO_2_-PbO-ZnO glass system was investigated. A series of 60TeO_2_-20PbO-(20-x)ZnO-xBaF_2_ glasses were prepared by melt quenching, in which x = 1, 2, 3, 5, 6, 7, and 9 mol%. Firstly, their structures and physical properties were studied. Then, ^57^Co (0.122 MeV), ^60^Co (1.173 and 1.332 MeV), ^137^Cs (0.662 MeV), ^133^Ba (0.081 and 0.356 MeV), and ^241^Am (0.059 MeV) were used as the radiation source to test each glass’s linear attenuation coefficient, the corresponding mass attenuation coefficient was calculated, and after which the results were compared with the results simulated by XCOM software. Then, the important radiation shielding parameters such as half-value layer (HVL), mean free path (MFP), and effective atomic number (Z_eff_) were evaluated. Finally, the neutron radiation-shielding characteristics are discussed by measuring the studied glass’s removal cross-section (RCS).

## 2. Materials and Methods

### 2.1. Sample Preparation

Firstly, 60TeO_2_-20PbO-(20-x)ZnO-xBaF_2_ glass was prepared using the melt quenching method. The purity of the raw materials is greater than or equal to 99.99%. The prepared samples were named TPZBF-1, -2, -3, -5, -6, -7 and -9 according to the content of BaF_2_. Each batch was melted in a pure Al_2_O_3_ ceramic crucible at 850 °C for 1.5 h and then cast onto a brass mold preheated at 200–250 °C. The formed glass block was then swiftly transferred to a muffle furnace at 250 °C for annealing until room temperature. Then, cooled glass block was grinded into a glass sheet with a thickness of about 0.75–1 mm, for which the cross-section of the glass sheet is 20 mm × 10 mm, and the two large surfaces of the glass were optically polished.

### 2.2. Radiation Shielding Experiment

The schematic and physical diagram of the device for measuring linear attenuation coefficient with physical radiation source is shown in Figure 1. The radiation source used was encapsulated in several 20 mm wide lead bricks of the device. The sample was placed 300 mm away from the radiation source. A collimator with a diameter of 5 mm was placed between the radiation source and the glass sample to resist the radiation scattered by the radiation source and reduce the pollution to the surrounding environment.

This study obtained the γ-ray attenuation coefficients in seven energy lines of the prepared barium-doped lead telluride glasses using a collimated narrow gamma-ray beam from the radioactive source described above.

To eliminate the influence of background radiation on the experimental results, the detector was used for thirty minutes without a radiation source before each experiment. The counting rate obtained without radioactive source is called background counting rate I_bg_. The random error can be reduced by averaging multiple experiments, whereby each radiation source irradiates each glass ten times, and the time of each experiment is five minutes. When the glass was placed, the average count rate is represented by the symbol I, and the count rate when the glass was not placed is represented by the symbol I_0_.

### 2.3. Other Physical Stability Experiments

The structure and physical properties of the TPZBF series glass were characterized as follows: According to Archimedes principle, the density was measured with pure water (ρ = 0.99980 g/cm^3^, 16 °C) as the normal temperature immersion liquid. The SEM of glass was carried out using a Fei quanta 650 scanning electron microscope under vacuum and acceleration of 500 kV. Because the scanned sample is non-conductive glass, it was necessary to spray gold on the sample before scanning. The XRD pattern of glass was obtained using the RIGAKU Ultima IV instrument, the rate was 0.02°/min, the experimental environment was room temperature, and the experimental range was 2θ = diffraction angle of 10–80°. The radiation source used in the experiment was Cu-Kalpha, with a wavelength of 0.154178 nm. The sample holder of the instrument was made of stainless steel, and the influence of the background was subtracted during the test.

## 3. Theory

According to the Lambert-Beer law, before and after the glass is placed, the counting rate obtained by the detector has the following relationship [21]:(1)$$\mu =1/t\times \ln\left[\left(I_{0}-I_{\mathrm{bg}}\right)/\left(I-I_{\mathrm{bg}}\right)\right]$$
where t is the thickness of the glass sample tested.

To eliminate the influence of density, the mass attenuation coefficient is introduced μ_m_. The following Formula (2) can calculate its value:(2)$$\mu_{m}=\mu /\rho$$
where ρ is the density of the material.

The half value layer HVL is the shielding material thickness that reduces the initial beam intensity to 1/2. It is a crucial shielding parameter connected to the thickness of the material, which the following Formula (3) can calculate [22]:(3)HVL=ln(2)/μ

The mean free path MFP is an important parameter that provides information about the distance a photon moves inside the glass sample. MFP can be obtained with the following Formula (4) [23]:(4)MFP=1/μ

The effective atomic number Z_eff_ values of the TPZBF glass systems were determined with the help of the Direct Method that was introduced by Manohara et al. [24].
(5)$$Z_{\mathrm{eff}}=(\sum_{i}f_{i}A_{i}\mu_{\mathrm{mi}})/(\sum_{i}f_{i}A_{i}\mu_{\mathrm{mi}}/Z_{i})$$
where f_i_ is the molar fraction, A_i_ is the atomic weight, and the Z_i_ is the atomic number.

The RCS values of each component glass were calculated to evaluate the neutron radiation shielding properties of TPZBF series glasses. The RCS plays a primary role in testing the medium’s ability to block the neutrons beam. In general, materials with high RCS values have greater protection against neutron radiation. The calculation process of RCS is shown in the following formulas [25,26].
(6)$$\mathrm{RCS}=\underset{i}{\sum }\rho_{i}(\sum^{\;}R/\rho {)}_{i}$$
where $\rho_{i}$ is the partial density of the i constituent and the $\sum^{\;}R/\rho$ the mass removal cross section.

In addition, since the physical radiation source used in this study can only provide energy in the range of 0.059–1.332 MeV, to better show the radiation shielding ability of the sample, XCOM software was used to estimate the mass attenuation coefficient of samples in the energy range of 0.05–15 MeV.

## 4. Results and Discussion

The prepared TPZBF glass is a series of transparent amorphous glass, of light yellow-green. After optical polishing the two sides of the glass, it was found that the surface of the TPZBF1-7 sample is transparent without crystallization. In contrast, the surface of the TPZBF-9 sample has a small number of minor crystallization points. The SEM image of the optically polished surface of the TPZBF-7 sample is shown in Figure 2a. It can be seen from the figure that the surface of TPZBF-7 is a glassy surface without crystal deposits, and the SEM images of the sample surface of TPZBF1-6 are all similar. An image of the defect position of the TPZBF-9 sample is shown in Figure 2b, which indicates that there is a crystal phase in the TPZBF-9 glass.

The XRD spectrum of TPZBF series glass is shown in Figure 3. It can be seen from the figure there are two broad humps between 2θ = 20–60° when x ≤ 7 mol%, and there are also no spikes. It indicates that the glass’s atomic arrangement is not in a long-range order, which confirms that TPZBF-1~TPZBF-7 glass samples are amorphous and have a stable structure. When x = 9 mol%, the crystallization peak appears in the spectrum, indicating crystals in the glass. Using card comparison, it was found that the crystal is Ba_2_ZnF_6_.

The composition, density and mass fraction of each element of TPZBF series glass are shown in Table 1. From TPZBF-1 to TPZBF-9, the density of the glass gradually increased from 6.243 g/cm^3^ to 6.327 g/cm^3^. The density of glass increases with an increase in BaF_2_ content. This is because the molar mass of BaF_2_ (175.32 g/mol) is higher than that of ZnO (81.38 g/mol). The density directly affects the radiation shielding material’s half-value layer and average free path. Generally speaking, the higher the density, the better the compactness of the glass and the stronger the radiation-shielding ability. High-density materials have more atoms and electrons per unit volume, which means that the material has a higher probability of interacting with gamma rays and can shield gamma rays better. This situation can also be seen from Equations (2) and (4), which show that the value of MFP is inversely proportional to density. The greater the density of the material, the smaller the MFP, which indicates that high-density materials have better radiation shielding performance when a high-energy ray pass through materials of equal thickness.

XCOM software was used to simulate the μ_m_ value of TPZBF in the photon energy range of 0.05–1.5 MeV which was compared with the experimental μ_m_ value to verify the accuracy of the experimental results. The experimentally obtained μ_m_ values and the XCOM simulated μ_m_ values are named (μ_m_) _Exp_ and (μ_m_) _XCOM_, respectively. A comparison between the two results is shown in Figure 4 below. It can be seen from the figure that the values of (μ_m_) _Exp_ and (μ_m_) _XCOM_ under different photon energies are close, and the changing trends are the same, which shows that the experimental results are accurate and reliable.

The relative difference Dev between the two can be obtained according to the following Formula (7):(7)$$\mathrm{Dev}=\left|[{\left(\mu_{m}\right)}_{\mathrm{Exp}}-{\left(\mu_{m}\right)}_{\mathrm{XCOM}}]/{\left(\mu_{m}\right)}_{\mathrm{XCOM}}\right|\times 100\%$$

The μ_m_ values of the two methods and their Dev are shown in Table 2. It is clear that the experimental results of lead tellurite glasses containing different concentrations of BaF2 are consistent with those of XCOM at all energies, which implies that the physical test results are in an agreement with the simulated by XCOM software.

From the data in the chart, it can be seen that the value of μ_m_ is closely related to the value of incident photon energy E_p_ and the content of BaF_2_ in the range of 0.059 MeV < E_p_ < 1.332 MeV.

It can be seen from the figure that the μ_m_ value decreased exponentially with the increase in E_p_ in the whole range. When BaF_2_ replaces the ZnO in the composition, the μ_m_ value increases accordingly. However, the degree of influence of the BaF_2_ content on the μ_m_ value is different under the condition of different E_p_. The BaF_2_ content has a more obvious impact on the μ_m_ value in the range of E_p_ < 0.662 MeV. Such as the μ_m_ value of TPZBF-1 is 4.9637 cm^2^/g, and the μ_m_ value of TPZBF-9 is 5.2108 cm^2^/g at E_p_ = 0.059 MeV, which is quite different, μ_m_ value is hardly affected by BaF_2_ content in the range of E_p_ ≥ 0.662 MeV. Similar to the value of TPZBF-1, the μ_m_ value is 0.0512 cm^2^/g, and the μ_m_ value of TPZBF-9 is 0.0513 cm^2^/g at E_p_ = 1.332 MeV, i.e., two are almost identical.

This phenomenon is because gamma photons interact with matter and lose most of their energy in collision with atoms. Therefore, different interaction modes will affect the results of radiation attenuation. When the energy of the gamma photon is lower than 2 MeV, the primary interaction modes are the main photoelectric effect and the Compton effect. For TPZBF series glass, the photoelectric effect mainly acts on the range of E_p_ < 0.662 MeV, while the Compton effect primarily acts on the range of E_p_ ≥ 0.662 MeV. The dependence of the photoelectric effect on the atomic number Z of the sample is higher than that of the Compton effect. As a heavy element, Ba has a higher atomic number than Zn, which increases the cross-section of the interaction between photons and glass. By improving the μ_m_ value in in the case of low E_p_, it is more affected by improving the component content of the heavy element.

Half Value Layer (HVL) and Mean Free Path (MFP) are valuable factors in predicting the photon protecting the proficiency of any material and to find the cutting amount of gamma radiation. The relationship between HVL value and photon energy and BaF_2_ content is shown in Figure 5. The HVL value of TPZBF series glass increases with photon energy and decreases slightly with the increase in BaF_2_ content, but the change of BaF_2_ content has no significant effect on the HVL value. For example, when the BaF_2_ content increases from 1% to 9% at E_p_ = 1.332 MeV, the HVL value of the glass decreases from 2.168 cm to 2.136 cm.

The mean free path of TPZBF series glasses was also studied. The MFP values of TPZBF-1, -7 and other tellurite glasses are shown in Figure 6. It should be noted that to reflect the radiation-shielding ability of the studied glass in more energy ranges, the MFP value in Figure 6 is not the experimental value but was obtained using theoretical calculation. As shown from Figure 6, the curves of TPZBF-1 and -7 almost coincide, and their MFP values are lower than other tellurite glasses, indicating that TPZBF series glasses are very effective at absorbing gamma rays and more effective than other tellurite glasses.

The effective atomic number Z_eff_ of TPZBF glass obtained by theoretical calculation, as shown in Figure 7, Z_eff_ is another critical factor in materials science and radiation physics, and it is an indispensable parameter in the study of gamma shielding. The Z_eff_ value reveals the radiation-proof ability of the material against gamma-ray attenuation when photons pass through the material. It is easier to intercept via collision with a large Z_eff_ atomic nucleus. They are highly attenuated in these materials. The high Z_eff_ value of the radiation-proof ability of glass shields against gamma radiation is better in practical shielding applications.

The Z_eff_ value of TPZBF glass increases with the increase in BaF_2_. In the TPZBF glasses, the highest Z_eff_ values occurred between 0.02 MeV and 0.03 MeV. This is because high Z elements, especially Pb elements, have a high μ_m_ value concentration at low E_p_, significantly promoting the photoelectric reaction. Moreover, an abrupt discontinuity can be seen in the µ/ρ curve due to the photoelectric effect near the absorption K-edge of the Te element at 0.0318 MeV. As E_p_ increases, the Z_eff_ of the glass gradually decreases because of the Compton scattering. Finally, the Z_eff_ value of the sample will gradually rise at the stage of E_p_>1.5MeV due to the influence of the electron pair effect..

The Z_eff_ values of the three glasses have little difference in the range of 0.2 MeV < E_p_ < 0.8 MeV, mainly due to the following two points. First, it can be seen from Formula (5) that the Z_eff_ value depends on the atomic number of each element and the product of the mass fraction and the μ_m_ value. Due to the fact that the atomic number and μ_m_ value of the Ba element are higher than that of the Zn element, as the Ba content increases, the Z_eff_ value also increases. Second, as the proportion of Ba in the composition increases, although the proportion of Pb is always 20%, the mass fraction of Pb decreases, which will lead to a decrease in the Z_eff_ value. As shown in Table 3, the value of (μ_m_)_Pb_/(μ_m_)_Ba_ is much greater than 1 in the range of 0.2 MeV < E_p_ < 0.8 MeV, while the value of (μ_m_)_Pb_/(μ_m_)_Ba_ gradually veers to 1 in the range of E_p_ > 0.8 MeV, so the three curves overlap in the range of 0.3 MeV < E_p_ < 0.8 MeV.

The RCS value of TPZBF series glass is shown in Figure 8. For further illustration, the RCS value of regular concrete and graphite are also presented in Figure 8. The RCS values are 0.10876, 10883, 0.10891,0.10913, 0.10918, 0.10919, and 0.10920 cm^−1^ for TPZBF-1~TPZBF-9, respectively. With the increase in BaF_2_, the RCS value of the glass increases. This phenomenon is because the Ba element with a higher atomic number replaces the Zn element with a lower atomic number in the composition so that the RCS value increases. However, the increasing trend gradually decreases. This is because Ba doping will lead Pb to decrease in the mass fraction with the largest atomic number, thereby reducing the increase in the RCS value. The RCS value of TPZBF-1 is the smallest of the TPZBF series glass. However, its RCS value is still much higher than regular concrete and graphite, indicating that TPZBF series glass has a more vital neutron-shielding ability than regular concrete and graphite.

## 5. Conclusions

In this investigation, the structural, optical and radiation safety characteristics of newly constructed lead tellurite glasses with 60TeO_2_-20PbO-(20-x)ZnO-xBaF_2_ were assessed. SEM and XRD spectra confirmed the amorphous nature of the samples. The physical radioactive source was used to test the glass μ_m_ value and simulations were performed with XCOM software to verify the accuracy of the measured data. Then, the HVL, MFP, Z_eff_, and RCS parameters of all selected glasses were calculated, which were with common commercial glass and graphite. The μ_m_, Z_eff_ and RCS value of glass increases and the HVL and MFP value decrease when BaF_2_ replaces ZnO. It is deduced that the addition of BaF_2_ to the studied glass system significantly improved the nuclear shielding qualities of prepared glasses.

## Figures and Tables

**Figure 1 materials-15-02117-f001:**
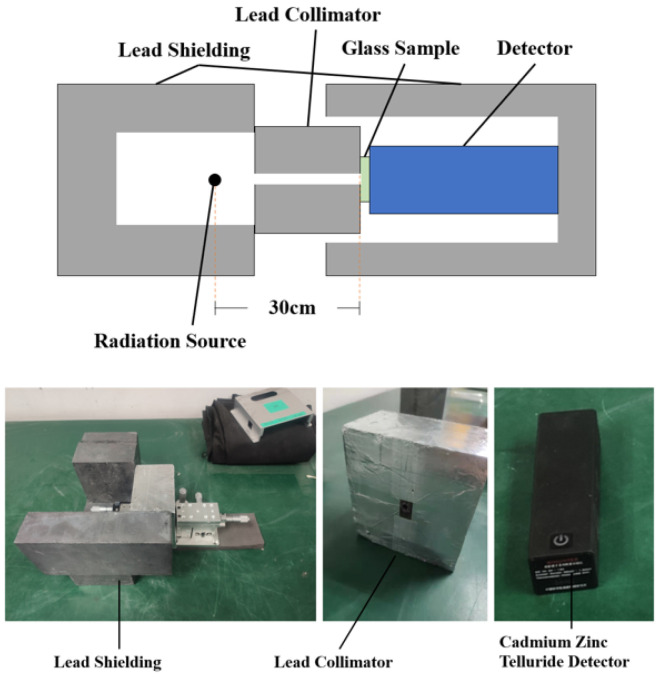
Structure diagram and physical diagram of the device for measuring linear attenuation coefficient with physical radiation source.

**Figure 2 materials-15-02117-f002:**
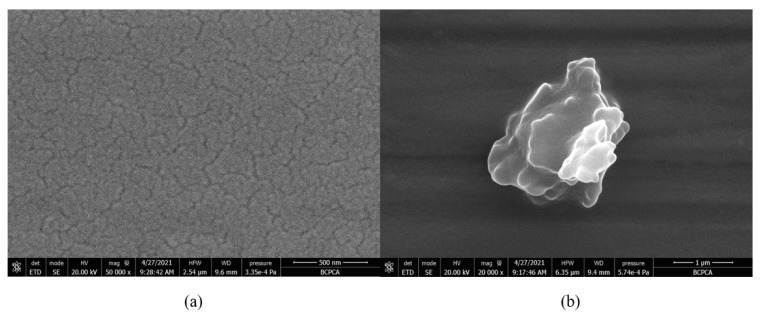
SEM micrographs of TPZBF-7 (**a**) and TPZBF-9 (**b**).

**Figure 3 materials-15-02117-f003:**
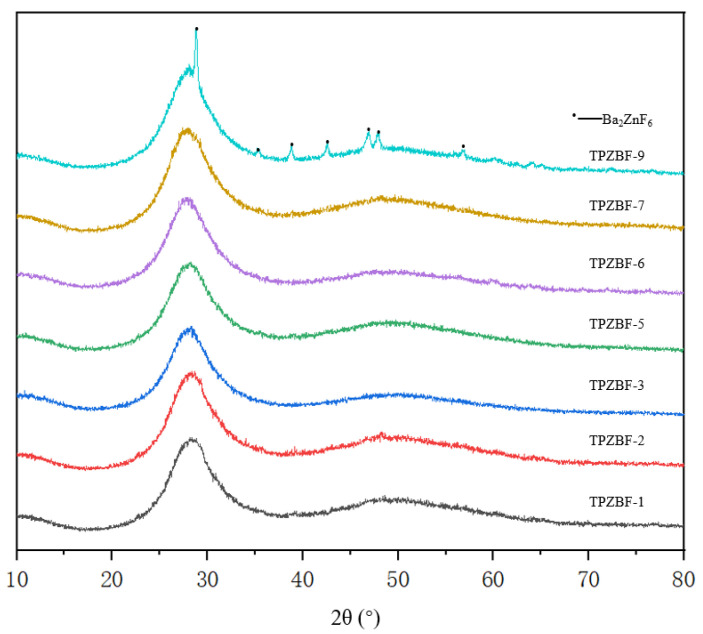
XRD diagram of TPZBF series glass.

**Figure 4 materials-15-02117-f004:**
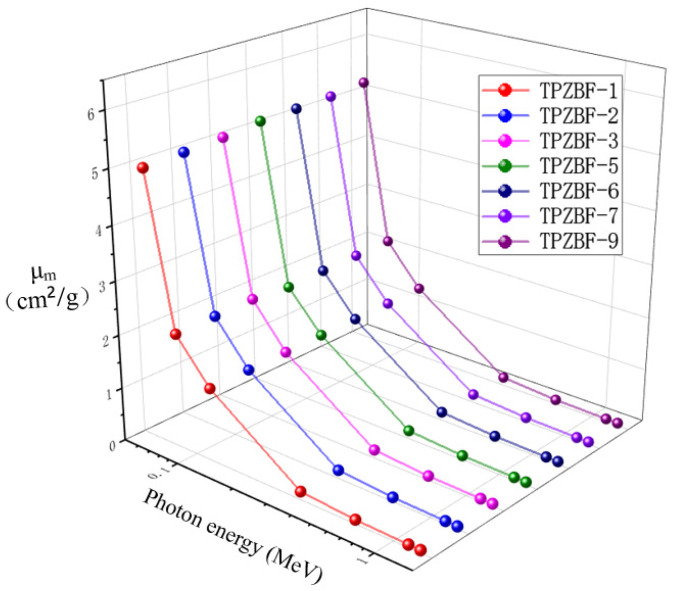
Comparison of mass attenuation coefficient obtained from experiment and XCOM simulation calculation.

**Figure 5 materials-15-02117-f005:**
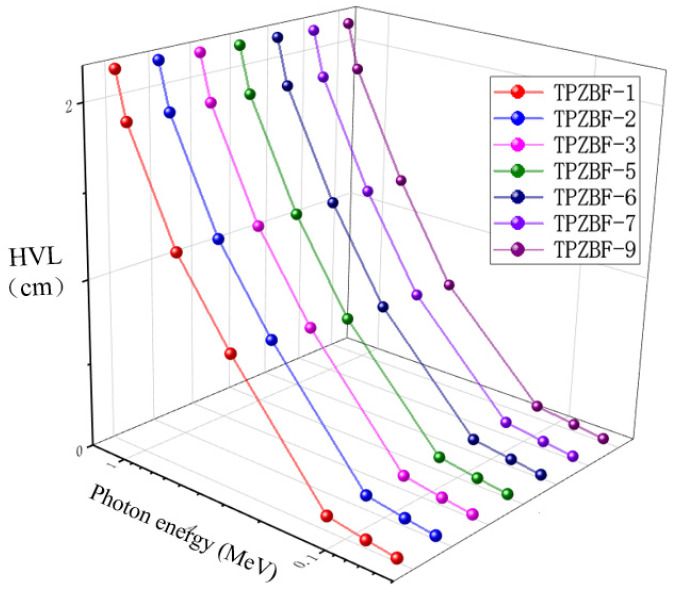
The relationship between HVL value and photon energy and BaF_2_ content.

**Figure 6 materials-15-02117-f006:**
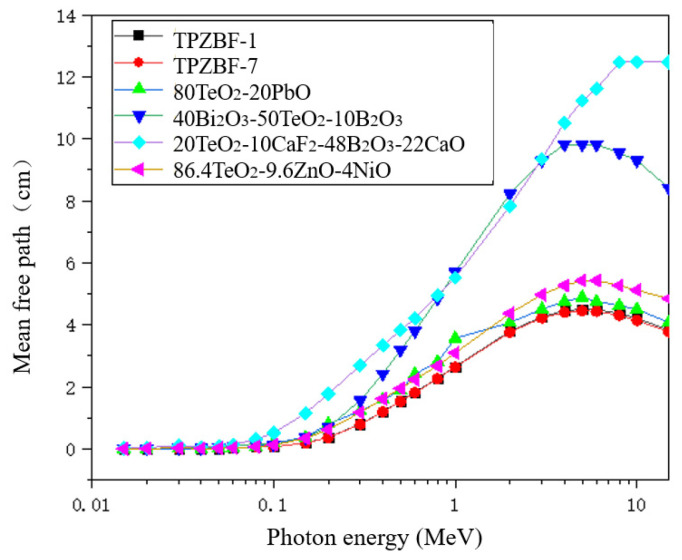
The relationship between MFP value and photon energy and BaF_2_ content.

**Figure 7 materials-15-02117-f007:**
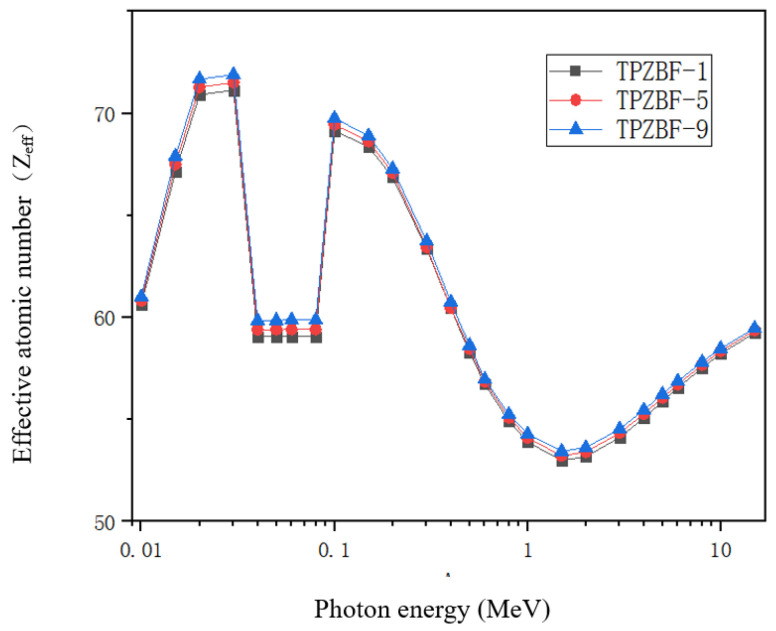
Z_eff_ value of TPZBF glass system.

**Figure 8 materials-15-02117-f008:**
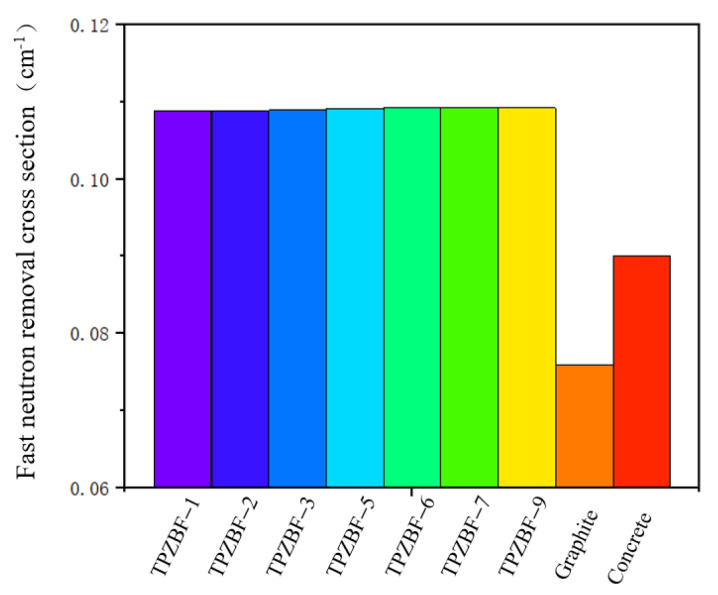
Removal cross section (RCS) of TPZBF series glass, regular concrete, and graphite.

**Table 1 materials-15-02117-t001:** Compositions and density of TPZBF series glass.

Sample Code	Chemical Composition				Density	Mass Fraction					
	TeO_2_	PbO	ZnO	BaF_2_	ρ/(g·cm^−3^)	O	F	Zn	Te	Ba	Pb
TPZBF1	60	20	19	1	6.243	0.16141	0.00241	0.07881	0.48574	0.00871	0.26292
TPZBF2	60	20	18	2	6.255	0.15944	0.00479	0.07422	0.48286	0.01732	0.26136
TPZBF3	60	20	17	3	6.267	0.15750	0.00715	0.06969	0.48002	0.02583	0.25982
TPZBF5	60	20	15	5	6.295	0.15368	0.01177	0.06077	0.47443	0.04255	0.25680
TPZBF6	60	20	14	6	6.306	0.15181	0.01405	0.05639	0.47168	0.05077	0.25531
TPZBF7	60	20	13	7	6.313	0.14995	0.01629	0.05206	0.46897	0.05889	0.25384
TPZBF9	60	20	11	9	6.327	0.14631	0.02071	0.04355	0.46363	0.07485	0.25095

**Table 2 materials-15-02117-t002:** Value and Dev of (μ_m_) _Exp_ and (μ_m_) _XCOM_.

Energy (MeV)		TPZBF-1	TPZBF-2	TPZBF-3	TPZBF-5	TPZBF-6	TPZBF-7	TPZBF-9
	Exp	4.9637	5.0006	5.0375	5.1100	5.1263	5.1482	5.2108
0.059	XCOM	4.9410	4.9790	5.0170	5.0920	5.1290	5.1650	5.2370
	Dev	0.46%	0.43%	0.41%	0.35%	0.05%	0.33%	0.50%
	Exp	2.1350	2.1532	2.1703	2.1045	2.1359	2.1532	2.1703
0.081	XCOM	2.1120	2.1280	2.1450	2.1760	2.1920	2.2080	2.2380
	Dev	1.09%	1.18%	1.18%	3.29%	2.56%	2.48%	3.03%
	Exp	1.3655	1.3712	1.3783	1.3846	1.3855	1.3912	1.3983
0.122	XCOM	1.3920	1.3930	1.3950	1.3970	1.3980	1.4000	1.4020
	Dev	1.90%	1.56%	1.20%	1.23%	0.89%	0.63%	0.26%
	Exp	0.1506	0.1533	0.1551	0.1542	0.1556	0.1563	0.1571
0.356	XCOM	0.1549	0.1549	0.1549	0.1549	0.1550	0.1550	0.1550
	Dev	2.78%	1.03%	0.13%	0.45%	0.39%	0.84%	1.35%
	Exp	0.0785	0.0785	0.0785	0.0785	0.0785	0.0786	0.0786
0.662	XCOM	0.0806	0.0807	0.0807	0.0808	0.0809	0.0809	0.0809
	Dev	2.61%	2.73%	2.70%	2.80%	2.94%	2.88%	2.87%
	Exp	0.0546	0.0547	0.0547	0.0547	0.0546	0.0547	0.0547
1.173	XCOM	0.0544	0.0544	0.0544	0.0545	0.0545	0.0545	0.0546
	Dev	0.46%	0.55%	0.50%	0.46%	0.17%	0.29%	0.26%
	Exp	0.0512	0.0512	0.0513	0.0513	0.0512	0.0512	0.0513
1.332	XCOM	0.0505	0.0506	0.0506	0.0506	0.0507	0.0507	0.0507
	Dev	1.35%	1.25%	1.40%	1.34%	1.05%	1.01%	1.14%

**Table 3 materials-15-02117-t003:** Mass attenuation coefficient μ_m_ (cm^2^/g) of Ba and Pb.

E_p_ (MeV)	0.2	0.3	0.4	0.5	0.6	0.8	1	1.5	2
Ba	0.4045	0.1891	0.1265	0.0992	0.0841	0.0674	0.0580	0.0459	0.0407
Pb	0.9985	0.4026	0.2323	0.1613	0.1248	0.0887	0.0710	0.0522	0.0460
Ba/Pb	2.468	2.212	1.836	1.625	1.483	1.315	1.224	1.137	1.129

## Data Availability

Not applicable.

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
