# Peer review of "Study on Radiation Shielding Properties of New Barium-Doped Zinc Tellurite Glass"

_materials, 2022, doi:10.3390/ma15062117_

Round 1
Reviewer 1 Report
- At introduction the aim main of present work should be brief and clear. Also, in abstract the same ideas are repeated.
- Why do select up to 9 mol% BaF2?
- Has a state diagram been set up for research samples at different concentration?
- The text should include one or two sentences about energy transfer.
- What is the temperature of the samples during irradiation?
- It should add some information about the chemical purity of samples (TeO2, PbO and ZnO).
- What is under of the inert gas were the samples heated and cooled?
- Avoid the use of pronoun. Eg We... (line 66) etc should be avoided.
- "In this study, using the collimated narrow gamma-ray beam from 57Co (0.122MeV), 80 60Co (1.173 and 1.332MeV), 137Cs (0.662MeV), 133Ba (0.081 and 0.356MeV) and, 241Am 81(0.059MeV) radioactive sources, the gamma-ray attenuation coefficient of the prepared 82 barium-doped lead telluride glass was obtained in seven energy lines" sentences are repeated in introduction and abstract.
- How did you separate the 60Co (1.173 and 1.332MeV) and 133Ba (0.081 and 0.356MeV) double energy lines?
- "The structure and thermal properties of TPZ F series glass were characterized as follows...page -3, line 92 .....“thermal properties” should be removed. Please consider your thermal properties. The texte should be revised. There are no studies on the "thermal properties".
- “Theoretical part” should be removed, the references must be given.
- Figure 2 should be shown in high resolution. For comparison in Fig.2 should be shown XRD patterns of Ba2ZnF6 sample.
- In Fig.3a and Fig.3b shows: pressure in Fig.3a (3.35e-4Pa) and Fig.3a (5.74e-4Pa), in Fig.3a BCPCA-500 nm, in Fig.3b BCPCA-1µ Does not constitute compatibility and the results are questionable.
- How do influence of gamma sources to lattice parameters and surface morphology of samples?
- The line 157-158 are sentences are repeated in lines 93-94.
- How do influence of gamma sources to density factors of samples?
- There is no difference between Tab.2 and Fig.4.
- The Fig.5 does not provide any graphical information.
- None of the experimental results were explained on the basis of a mechanism. It should be revised again.
- There is no information about the color centers formed after gamma radiation in optical properties and mechanisms (migration, recombination... et al.,).
- Why do the attenuation coefficient values of the samples increase as the increasing gamma absorption dose decreases? What mechanisms can explain these processes? Please explain.
Author Response
Dear Reviewer:
I'm glad you reviewed the manuscript. You have provided me with many valuable comments, and the following are responses to your comments.
- At introduction the aim main of present work should be brief and clear. Also, in abstract the same ideas are repeated.
Reply:
We have rewritten the Abstract and Introduction as you suggested.
- Why do select up to 9 mol% BaF2?
Reply:
TeO2 can form glass with either PbO or BaO. Still, when the three are mixed, they cannot form glass. Glass network formers such as B2O3, P2O5, or ZnO must be required to incorporate PbO and BaO into a tellurate glass matrix obtain a glass with a high refractive index and density with relatively low toxicity. In this study, we choose 60TeO2-20PbO-20ZnO as the matrix, gradually add BaF2 to it, and reduce the content of ZnO as a glass network former. The highest selection of BaF2 is 9% because only so much BaF2 and PbO can coexist at most. Amorphous glass cannot be prepared when the BaF2 content is greater than 9%, and only pale yellow-green ceramics can be obtained.
- Has a state diagram been set up for research samples at different concentration?
Reply:
We believe that the focus of this study is on the radiation shielding ability of the glass system, not on the molding range of the glass, so we did not draw a state diagram to avoid distraction, other properties of glass. You can check our previous research https://doi.org/ 10.1155/2021/6466344.
- The text should include one or two sentences about energy transfer.
Reply:
Sorry, although the subject of this study is glass, its fluorescence spectrum has not been tested, so energy transfer cannot be discussed,
- What is the temperature of the samples during irradiation?
Reply:
The tests in this study were all performed at room temperature (approximately 20°C).
- It should add some information about the chemical purity of samples (TeO2, PbO and ZnO).
Reply:
The purity of all raw materials is ≥99.99%, the writing in the original manuscript seems ambiguous, and we have revised this.
- What is under of the inert gas were the samples heated and cooled?
Reply:
The samples were prepared in an air environment, and no inert gas was used, so we did not mention it in the manuscript.
- Avoid the use of pronoun. Eg We... (line 66) etc should be avoided.
Reply:
Thanks for your reminder, we have replaced all occurrences of the pronoun "We" in the manuscript.
- "In this study, using the collimated narrow gamma-ray beam from 57Co (0.122MeV), 80 60Co (1.173 and 1.332MeV), 137Cs (0.662MeV), 133Ba (0.081 and 0.356MeV) and, 241Am 81(0.059MeV) radioactive sources, the gamma-ray attenuation coefficient of the prepared 82 barium-doped lead telluride glass was obtained in seven energy lines" sentences are repeated in introduction and abstract.
Reply:
We have revised the Abstract and Introduction.
- How did you separate the 60Co (1.173 and 1.332MeV) and 133Ba (0.081 and 0.356MeV) double energy lines?
Reply:
The detector we used has an accuracy of 1keV and can read out counts at every keV. For the multiple energy lines of the same radioactive source, we choose the more accurate lines and have less mutual interference to count. For example, 133Ba has four energy lines of 0.081, 0.278, 0.302, and 0.356MeV, and we chose the two farthest line 0.081 and 0.356MeV. For the overlapping part, we use the method of nonlinear fitting to construct a function and compare it with the results of the XCOM simulation to find the most similar set of functions.
- "The structure and thermal properties of TPZ F series glass were characterized as follows...page -3, line 92 ......“thermal properties” should be removed. Please consider your thermal properties. The text should be revised. There are no studies on the "thermal properties".
Reply:
We have modified the description of this error.
- “Theoretical part” should be removed, the references must be given.
Reply:
We have removed unsubstantiated descriptions
- Figure 2 should be shown in high resolution. For comparison in Fig.2 should be shown XRD patterns of Ba2ZnF6 sample.
Reply:
We have replaced the image with a higher resolution.
- In Fig.3a and Fig.3b shows: pressure in Fig.3a (3.35e-4Pa) and Fig.3a (5.74e-4Pa), in Fig.3a BCPCA-500 nm, in Fig.3b BCPCA-1µ Does not constitute compatibility and the results are questionable.
Reply:
There is no problem with the pictures. The pressure difference between the two is to scan the amorphous smooth glass surface, and the other is to scan the crystallization point. To obtain their respective clear images, different voltages are used. From the time in the picture, it can be seen that the two photos are not taken simultaneously, and the instrument has been debugged in the middle.
- How do influence of gamma sources to lattice parameters and surface morphology of samples?
Reply:
Our sample is mainly amorphous glass, and the only TPZBF-9 sample contains a small amount of crystalline phase, so we have not studied the lattice parameters. As for the surface morphology, it is a pity that we only have the SEM image before the irradiation experiment. After receiving your comments, we tried to contact the laboratory for the SEM test, but due to the epidemic's impact, we were unable to reproduce the SEM test in the end.
- The line 157-158 are sentences are repeated in lines 93-94.
Reply:
We have removed this duplicate sentence.
- How do influence of gamma sources to density factors of samples?
Reply:
We measured the density of each glass sample before and after the experiment (each sample was tested six times and averaged), and the results did not change significantly. This may be due to the small measurement of the radioactive source used in our experiment or the short irradiation time, or it may be due to the stability of the glass itself. Since the exact reason cannot be determined, we have not mentioned this aspect in the text.
- There is no difference between Tab.2 and Fig.4.
Reply:
Thanks for your suggestion, our original idea was to present the accuracy and trend of data through different dimensions. We have now replaced Figure 4 with a 3D dot-and-line plot summarizing all the data to highlight the trend of μm as a function of incident energy and BaF2 content.
- The Fig.5 does not provide any graphical information.
Reply:
This is our mistake. The original figure 5 cannot show the change trend well, so we have redrawn another type of HVL diagram (three-dimensional dotted line diagram) to better show the change trend of HVL value with the change of glass composition and incident energy.
- None of the experimental results were explained on the basis of a mechanism. It should be revised again.
Reply:
We're sorry, but our ability does not allow us to make changes based on your comment.
- There is no information about the color centers formed after gamma radiation in optical properties and mechanisms (migration, recombination... et al.,).
Reply:
We're sorry, but our ability does not allow us to make changes based on your comment.
- Why do the attenuation coefficient values of the samples increase as the increasing gamma absorption dose decreases? What mechanisms can explain these processes? Please explain.
Reply:
We're sorry, but our ability does not allow us to make changes based on your comment.
In addition, we have retouched the English language and style of the manuscript. Thank you for your review comments.
Kind regards.
Yours sincerely,
Shiyu Yin
Reviewer 2 Report
The reviewed work is devoted to Pb,Zn,Ba,-tellurite glass with different content of BaF2. Authors study the radiation attenuation characteristics of the glass irradiated gamma sources.
As the authors write in the introduction there is need to obtain low-cost and non-toxic radiation shielding materials. Unfortunately, the proposed glass is neither cheap (TeO2 is expensive) nor non-toxic (contains PbO). The latter inconvenience can be neglected bearing in mind that crystal glass containing more than 20% PbO is widely produced. However, the testing of new materials is worth attention.
Some commands and correction are needed as following:
- 1. The distance of 30 cm is incorrectly labeled in the fig.
- The references should be rework, eg. page 4, line 113: Manohara et al. [19]. there are other authors under the number [19].
- 2 is of poor quality. There is no need to put (a.u.) when it is no scale.
- XRD patterns cannot be verified without providing wavelength of the x-ray tube.
- Page 5. Lines 151,152. There is no evidence that the authors would claim that TPZBF-9 glass-ceramics will have lower mechanical properties. Some glass-ceramics has higher strength, see Corning or Schott glass-ceramics.
- Page 5, Lines 161,162. On what basis the authors write that the structure becomes denser due to the increase of the bond ionicity. ZnO bond is stronger than BaF2 what is manifested by higher melting temperature of the former.
- The origin of the results for higher then 1.3 MeV of photon energy presented in the figures 6 and 7 is unclear due to the radioactive sources used.
Author Response
Dear Reviewer:
I am glad that you have reviewed the manuscript. The following is the response to your review comments.
- The distance of 30 cm is incorrectly labeled in the fig.
Reply:
I'm sorry for this low-level error in text editing. I've modified it and replaced it with the correct diagram.
- The references should be rework, eg. page 4, line 113: Manohara et al. [19]. there are other authors under the number [19].
Reply:
Here we are citing the wrong literature, which I have corrected.
- 2 is of poor quality. There is no need to put (a.u.) when it is no scale. XRD patterns cannot be verified without providing wavelength of the x-ray tube.
Reply:
I have corrected these errors and rewrote this section, as detailed in response to the fourth comment.
- Page 5. Lines 151,152. There is no evidence that the authors would claim that TPZBF-9 glass-ceramics will have lower mechanical properties. Some glass-ceramics has higher strength, see Corning or Schott glass-ceramics.
Reply:
Please allow me to explain in detail here. In the original chapter 4.1, there was a problem with our expression logic. What we really mean is:
We have prepared TPZBF series glass, in which the content of BaF2 is 1% - 11%. What we want is a series of transparent amorphous glass. Theoretically, it should be glass with little or no crystallization rather than the glass-ceramics obtained by deliberately adding a nucleating agent.
TPZBF1-9 is transparent glass, but the TPZBF-9 glass has tiny crystallization points that are difficult to observe with the naked eye. As for tpzbf-11, it is opaque light green ceramic (for this reason, tpzbf-11 is not mentioned in the manuscript).
After that, through the XRD test, we know that when the content of BaF2 is 1% - 7%, the glass is always amorphous, and the crystal peak appears in the XRD image of TPZBF-9 glass. Then we carried out further SEM and XRD tests on TPZBF-9, and finally confirmed that most of the structure of TPZBF -9 glass is amorphous, and there is a crystal phase only at the crystallization point.
In addition, in the process of processing our glass samples, TPZBF1-7 glass needs to be cut repeatedly with a glass knife before it can be cut into small pieces from a large block. In contrast, TPZBF-9 glass can be broken directly by hand at the position with crystallization point. Therefore, we believe that the structure of tpzbf-9 glass at the crystallization point is mutated, which will lead to the decline of the overall mechanical strength. However, we have no equipment to test the specific mechanical strength of the glass, therefore, we only covered this part in one sentence. I am sorry for the confusion it has caused you. In addition, in our research on the optical properties of TPZBF series glass, TPZBF-9 glass also mutated compared with other glasses, resulting in the decline of performance. For details, you can refer to our article: https://doi.org/10.1155/2021/6466344.
When we first wrote the manuscript, because we wanted to focus on the radiation shielding ability of glass, the chapter on the structure was too brief and not logical. We apologize for this. Ultimately, we decided to rewrite this chapter after discussion. The new manuscript addresses only the effect of glass composition on structure, omitting the unprovable powerless conclusions..
- Page 5, Lines 161,162. On what basis the authors write that the structure becomes denser due to the increase of the bond ionicity. ZnO bond is stronger than BaF2 what is manifested by higher melting temperature of the former.
Reply:
I've removed the erroneous sentence.
- The origin of the results for higher then 1.3 MeV of photon energy presented in the figures 6 and 7 is unclear due to the radioactive sources used.
Reply:
The MFP value shown in Fig. 6 and the Zeff value shown in Fig. 7 are based on a theoretical calculation to better describe the shielding characteristics of glass materials. Because the physical radiation source we use can only provide incident energy in the range of 0.059 to 1.332 MeV, a larger incident energy range can be used in the theoretical calculation, such as 0 to 15 MeV.
I added this to the description of Figures 6 and 7.
- As the authors write in the introduction there is need to obtain low-cost and non-toxic radiation shielding materials. Unfortunately, the proposed glass is neither cheap (TeO2 is expensive) nor non-toxic (contains PbO). The latter inconvenience can be neglected bearing in mind that crystal glass containing more than 20% PbO is widely produced. However, the testing of new materials is worth attention.
We used lead telluride glass as our research object because it was found in our previous study (DOI number: https://doi.org/10.1155/2021/6466344) that lead telluride glass has an extremely high refractive index (at 650nm, refractive index n>2.0) and nice optical performance. For optical components, the higher the refractive index, the thinner the material can be, and the smaller the volume of the entire optical system. In some special working environments such as space operations and medical devices, the advantages of high refractive index glass are undeniable, so we choose lead telluride glass as the glass matrix.
We did not reflect this in the original manuscript, and thanks for your reminder. We have revised the Introduction section.
In addition, we have retouched the English language and style of the manuscript. Thank you for your review comments.
Kind regards.
Yours sincerely,
Shiyu Yin
Reviewer 3 Report
- Why the authors used PbO and TeO2, however PbO has a toxic effect and TeO2 is very cost
- Please extent the introduction and consider the following: -the importance of using radiation shielding, -Why glasses for radiation shielding? the importance of your glass systems and the novelty in this work
- I think this is not correct (The theoretical calculation of the Zeff value is carried out using the well-known XCOM database.) since XCOM can only calculate the MAC
- Please check the subscript (between 10−3 and 105 MeV. )
- In section 4.3: WinXCOM software was....please check XCOM or WinXcom, since some time you use XCOM and sometime WinXcom
- In section 4.2, I found equation for MFP (equation 6), however you already mentioned this equation as equation 4 in section 3, please check
- section 4.3: please check the number of equation 6, i think this is equ 7
- In Fig.6, I fund a peak in MFP at high energy for one sample, why this peak appears?
- some recent Refs can be added to the introduction to improve the current work. For example you can refer to the following papers:
Journal of Non-Crystalline Solids 535 (2020) 119993
Materials Research Bulletin 142 (2021) 111383
Ceramics International 45 (2019) 24858–24864
Author Response
Dear Reviewer:
I am glad that you have reviewed the manuscript. The following is the response to your review comments.
- Why the authors used PbO and TeO2, however PbO has a toxic effect and TeO2 is very cost
Reply:
We used lead telluride glass as our research object because it was found in our previous study (DOI number: https://doi.org/10.1155/2021/6466344) that lead telluride glass has an extremely high refractive index (at 650nm, refractive index n>2.0) and nice optical performance. For optical components, the higher the refractive index, the thinner the material can be, and the smaller the volume of the entire optical system. In some special working environments such as space operations and medical devices, the advantages of high refractive index glass are undeniable, so we choose lead telluride glass as the glass matrix.
We did not reflect this in the original manuscript, and thanks for your reminder. We have revised the Introduction section.
- Please extent the introduction and consider the following: -the importance of using radiation shielding, -Why glasses for radiation shielding? the importance of your glass systems and the novelty in this work
Reply:
Thanks for your valuable ideas. We have rewritten the introductory part of the manuscript.
- I think this is not correct (The theoretical calculation of the Zeff value is carried out using the well-known XCOM database.) since XCOM can only calculate the MAC
Reply:
Thanks for pointing this out. We have removed these erroneous sentences.
- Please check the subscript (between 10−3 and 105 MeV. )
Reply:
Thanks for pointing it out. We have fixed the error. Sorry for such a low-level text editing mistake.
- In section 4.3: WinXCOM software was....please check XCOM or WinXcom, since some time you use XCOM and sometime WinXcom
Reply:
Thanks for pointing out we have unified the "WinXCOM" and "XCOM" that appear in the article as "XCOM."
- In section 4.2, I found equation for MFP (equation 6), however you already mentioned this equation as equation 4 in section 3, please check
Reply:
We have removed this recurring formula and revised the text description.
- section 4.3: please check the number of equation 6, i think this is equ 7
Reply:
Thanks for pointing it out. We have fixed the error. Sorry for such a low-level text editing mistake.
- In Fig.6, I fund a peak in MFP at high energy for one sample, why this peak appears?
Reply:
Sorry, this spike is due to a low-level error. The curve in the figure should be smooth. We have modified it and replaced it with the correct picture. This error is because the MFP value of the sample is not given in the literature we refer to. We use the m value of the sample given in the paper to calculate the MFP value. In the calculation process, we mistakenly write "0.034" as "0.024" so that the final calculated MFP value changes from "5.270" to "7.467". The data cited are from the references of this manuscript [12], and its DOI number is https://doi.org/10.1016/j.ceramint.2020.04.240
- some recent Refs can be added to the introduction to improve the current work. For example you can refer to the following papers:
Journal of Non-Crystalline Solids 535 (2020) 119993
Materials Research Bulletin 142 (2021) 111383
Ceramics International 45 (2019) 24858–24864
Reply:
Thank you for these references, which fit my article well and we have added them as a reference to this manuscript.
In addition, we have retouched the English language and style of the manuscript. Thank you for your review comments.
Kind regards.
Yours sincerely,
Shiyu Yin
Round 2
Reviewer 1 Report
The authors have addressed the reviewer's comments in the revised manuscript. I suggest its publication.
Author Response
Dear Reviewer:
We are submitting a new manuscript this time because the academic editor reminded us that we need to add the detailed conditions of the XRD test, which are added in lines 107-109 of the latest manuscript.
Thank you again for your careful review and valuable comments on our manuscript. Your comments have helped us improve the quality of the manuscript.
Kind regards.
Yours sincerely,
Shiyu Yin
Reviewer 3 Report
The authors improved the paper
Author Response

(The authors gave the same response as above.)
